# Scalable Boltzmann generators for equilibrium sampling of large-scale materials

Maximilian Schebek ®[1] ✉, Frank Noé ®[1,2,3,4] & Jutta Rogal ®[1,5]

Generating equilibrium ensembles of structures is essential for modeling molecules and materials, yet traditional simulators like molecular dynamics suffer from limited sampling efficiency. Boltzmann Generators introduced the concept of one-shot deep learning for equilibrium sampling, but scalability to large systems has remained a major challenge. Here, we overcome this scaling limitation with a Boltzmann Generator architecture that can model large materials systems. Our approach combines augmented coupling flows with graph neural networks to exploit local environments, enabling energy-based training and rapid inference. Compared to previous designs, it trains faster, uses fewer resources, and achieves superior sampling efficiency. Crucially, it transfers to much larger system sizes, allowing efficient sampling of materials with simulation cells exceeding a thousand atoms. We demonstrate its capabilities on Lennard-Jones crystals, mW water ice phases, and the silicon phase diagram, producing accurate equilibrium ensembles and free energies across scales where finite-size effects vanish.

Generative models are emerging as a promising alternative to traditional sampling techniques such as molecular dynamics (MD) and Monte Carlo (MC) simulations for generating equilibrium ensembles of many-particle systems. Unlike MD or MC, which explore the configuration space through sequential updates over time[1,2], generative models aim to learn a direct mapping from a base distribution, providing immediate access to uncorrelated samples from the equilibrium distribution of interest. The key advantage of such a model is that it enables one-shot generation of statistically independent equilibrium samples, bypassing the long correlation times and slow mixing inherent to conventional methods. Consequently, generative models have the potential to significantly reduce the cost of molecular simulations, particularly for computing thermodynamic ensemble averages and free energies. Building on the work by Noé et al. on Boltzmann Generators (BGs)[3], the combination of generative models such as normalizing flows[4,5] with statistical reweighting has been applied across diverse areas within molecular and condensed matter sciences. Examples include direct generation of equilibrium configurations from random noise and transformation of simulated distributions to explore different potentials or thermodynamic conditions, with

applications ranging from proteins[6–9] and small molecules[10–12] to condensed phase systems such as liquids[13,14], atomic solids[15–18], and molecular crystals[19,20].

Despite their promising properties, scaling BGs to large system sizes remains a significant challenge due to their high training costs and low sampling efficiencies. Consequently, most of the above examples have focused on proof-of-concept problems, such as Lennard-Jones clusters and small molecules with fewer than 100 atoms. In the context of materials science, the largest systems studied to date are atomic solids comprising approximately 500 particles. Yet, these models have demonstrated very low sampling efficiencies despite requiring nearly one GPU-year of training[15]. As a result, their computational cost exceeds that of MD-based simulations by several orders of magnitude, rendering them impractical for applications to realistic systems. This limitation is particularly critical in condensed-phase systems, where far larger system sizes are crucial to ensure statistically converged and physically meaningful predictions[21–23]. Recent work has therefore focused on scaling BGs to larger systems by developing improved architectures capable of handling substantially increased system sizes and on amortizing training costs through

[1]Department of Physics, Freie Universität Berlin, Berlin, Germany. [2]Department of Mathematics and Computer Science, Freie Universität Berlin, Berlin, Germany. [3]Microsoft Research AI for Science, Berlin, Germany. [4]Department of Chemistry, Rice University, Houston, TX, USA. [5]Initiative for Computational Catalysis, Flatiron Institute, New York, NY, USA. ✉e-mail: m.schebek@fu-berlin.de

transferable models[18,24,25]. Although initial progress has been made, extending BGs to systems with thousands of atoms remains beyond current capabilities.

In this work, we address this challenge by developing BGs that scale to large crystalline materials. In contrast to previous approaches targeting condensed-phase systems[15,16,20], we adopt a locality assumption and base the generative process on local environments rather than the full configuration. This is achieved by leveraging graph neural networks[26] in combination with augmented coupling flows[27], which enable energy-based training without requiring samples from the target distribution and remain computationally efficient during inference. We show that this local architecture not only allows to train models to far higher sampling efficiencies but also significantly reduces the training costs. A crucial feature of our architecture is its transferability across system sizes, thereby allowing models trained on small systems to be employed on much larger ones, which drastically reduces the computational cost compared to a direct training on large systems. Conditioning the model on external parameters, such as thermodynamic states or the atomic interaction potential, further amortizes the training costs. We demonstrate the potential of our approach by producing accurate equilibrium ensembles for a diverse set of systems, including crystalline phases of the Lennard-Jones potential as well as various parameterizations of the Stillinger-Weber potential[28] spanning monatomic water, germanium and silicon, and reporting accurate Gibbs and Helmholtz free energies for system sizes well above 1000 atoms.

## Results

### Scaling Boltzmann generators to large systems

Central to scaling any architecture in the context of many-body systems is formulating the learning problem based on local structural features[29,30]. Following this principle, we aim to train a flow to learn coordinate transformations based on local environments, allowing for efficient training on small systems and seamless transfer to larger ones. While in the context of continuous flows, configurational updates can be defined based on local structural environments[8,31], evaluating the densities of these models requires computing the Jacobian trace, which scales poorly to high dimensions[32–34]. Another drawback is that training continuous flows requires samples from the target distribution[32], which are, in fact, the very quantities the method aims to generate. Although methods for reducing the cost of the Jacobian trace evaluation[33,35] and for continuous models that do not require target samples[36,37] are active areas of research with some initial progress, these approaches are not yet mature enough to scale to the system sizes relevant to this work. For this reason, we instead focus on coupling flows[38,39]. These architectures support efficient density estimation by partitioning the input into two channels according to $\mathbf{x} = (\mathbf{x}_A, \mathbf{x}_B)$ and updating one conditioned on the other via $\mathbf{x}'_A = g(\mathbf{x}_A | C(\mathbf{x}_B))$, were $g$ is a bijection parametrized by a conditioner $C$, producing a triangular Jacobian that can be evaluated analytically. A crucial feature of coupling flows is their compatibility with likelihood-based optimization schemes, allowing the model to be trained using only the potential energy function of the target distribution, without requiring any samples from it.

As coupling flows need to split coordinates either across particles or across spatial dimensions or a combination thereof, the conditioner $C$ does not have access to complete three-dimensional interatomic distances, as its input can only consist of a subset of the coordinates[40] (see Supplementary Note 1 for a detailed discussion on the limitations of standard coupling flows). Consequently, previous architectures often rely on the absolute coordinates of all atoms in the system[15,19], and we will refer to this approach as a 'global' architecture in the following. To overcome this architectural limitation of coupling flows, we leverage the augmented coupling flow framework[27]. Augmented flows introduce auxiliary variables $\mathbf{a} \in \mathbb{R}^{3N}$ and enable the splitting to

be performed between physical and auxiliary variables, thereby updating the physical variables $\mathbf{x} \in \mathbb{R}^{3N}$ conditioned on $\mathbf{a}$ and vice versa. Importantly, this scheme retains full three-dimensional coordinates within the physical and auxiliary space and, thus, allows for computing interatomic distances. We model the auxiliary system as a noised copy of the physical system[41] and define the joint base distribution of the augmented system as

$$q(\mathbf{x}, \mathbf{a}) = q(\mathbf{x}) \mathcal{N}(\mathbf{a}; \mathbf{x}, \eta_q^2 \mathbf{I}) . \tag{1}$$

Here, $q(\mathbf{x})$ is the base distribution of the physical variables and $\mathcal{N}(\cdot; \mathbf{x}, \eta_q^2 \mathbf{I})$ denotes the normal distribution centered at $\mathbf{x}$ with covariance matrix $\eta_q^2 \mathbf{I}$, where $\mathbf{I}$ is the identity matrix and $\eta_q \in \mathbb{R}_{>0}$. In this construction, the auxiliary system retains structural information of the physical variables that can be exploited by the flow transformation.

Utilizing the auxiliary variables, we define the flow transformation for particle $i$ by

$$\mathbf{x}'_i = g(\mathbf{x}_i | \mathbf{h}_i^{\mathbf{a}}) , \tag{2}$$

where $g$ is parametrized by the embedding $\mathbf{h}_i^{\mathbf{a}}$, which is computed using a graph neural network (GNN)[26] that aggregates information from each auxiliary particle's local neighborhood (see Fig. 1). Details on the implementation can be found in Supplementary Note 3. A crucial feature of the local approach presented here is its linear scaling with system size, achieved by fixing the number of neighbors. This stands in stark contrast to global architectures based on, for example, attention mechanisms, which learn interactions between all particles, resulting in quadratic scaling with system size[15,42].

To ensure that the bijector, which we implement as rational quadratic splines[43], remains applicable to larger system sizes, it is crucial that the magnitude of its input does not scale with the system size. One possibility that we follow in our proposed architecture is to model the displacements from the ideal crystal lattice rather than absolute positions. This makes the input size-independent and preserves the model's ability to generalize to arbitrary system sizes. Accordingly, we define the physical base distribution as $q(\mathbf{x}) = \mathcal{N}(\mathbf{x}; 0, \eta_q^2 \mathbf{I})$.

For a flow acting on the augmented space, the change-of-variable for the augmented system writes

$$q_\theta(\mathbf{x}', \mathbf{a}') = q(\mathbf{x}, \mathbf{a}) \left| \det J_{\mathcal{F}_\theta}(\mathbf{x}, \mathbf{a}) \right|^{-1} , \tag{3}$$

where $\det J_{\mathcal{F}_\theta}$ denotes the Jacobian determinant of the flow transformation $\mathcal{F}_\theta$. The target distribution of the auxiliary variables is $\pi(\mathbf{a}|\mathbf{x}) = \mathcal{N}(\mathbf{a}; \mathbf{x}, \eta_p^2 \mathbf{I})$, such that the flow is trained to optimize the joint target distribution

$$p(\mathbf{x}, \mathbf{a}) = p(\mathbf{x}) \mathcal{N}(\mathbf{a}; \mathbf{x}, \eta_p^2 \mathbf{I}) , \tag{4}$$

where $p(\mathbf{x})$ is the equilibrium distribution of the ensemble of interest. Here, we focus on the canonical (or *NVT*) ensemble, whose equilibrium distribution is the Boltzmann distribution. The flow is trained by minimizing the Kullback-Leibler (KL) divergence between the generated and target distributions in the joint space. More details on the canonical ensemble and the explicit form of the loss function can be found in "Methods".

Within the augmented setting, only the joint generated density $q_\theta(\mathbf{x}', \mathbf{a}')$ can be evaluated exactly. However, the marginal of the physical system can be obtained by integrating out the auxiliary degrees of

**(a)**

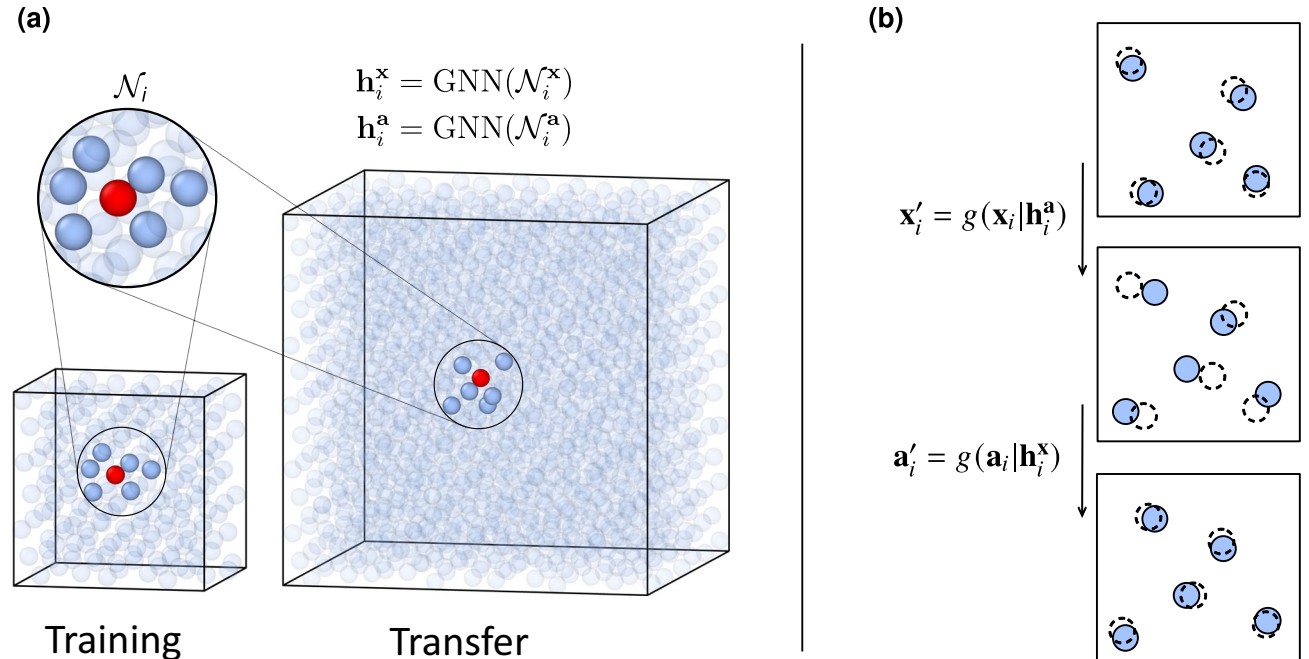

**(b)**

**Fig. 1 | Size-transferable augmented flow. a** A GNN learns environment-dependent particle embeddings, $\mathbf{h}_i$, in a small system and is transferable across system size via each particle's local neighborhood, $\mathcal{N}_i$. **b** Physical and auxiliary particles are updated sequentially using the learned embeddings. In each step, either all auxiliary particles or all physical variables are updated.

freedom, which can be approximated as[41]

$$q_\theta(\mathbf{x}') \approx \frac{1}{M} \sum_{m=1}^{M} \frac{q_\theta(\mathbf{x}', \mathbf{a}_m)}{\pi(\mathbf{a}_m | \mathbf{x}')}, \qquad (5)$$

with $\mathbf{a}_m \sim \pi(\cdot | \mathbf{x}')$ and $M$ denoting the number of auxiliary samples drawn per generated physical sample. Unless stated otherwise, all results obtained from the marginal densities presented in this work were obtained using $M = 200$, for which the free energy estimates from the marginal density were found to be converged (see Supplementary Note 1). Importantly, while the evaluation of the marginal distribution does require $M$ additional inverse passes through the flow per generated sample (since $q_\theta(\mathbf{x}', \mathbf{a}') = q(\mathcal{F}_\theta^{-1}(\mathbf{x}', \mathbf{a}'))|\det J_{\mathcal{F}_\theta^{-1}}(\mathbf{x}', \mathbf{a}')|$), it does not require additional evaluations of the target potential.

As detailed in "Methods", the calculation of the (reduced) Helmholtz free energy associated to a distribution $\mu$, $f_\mu = F_\mu / k_B T_\mu = -\log Z_\mu$, requires an estimate of the partition function $Z_\mu$. While this quantity cannot be computed directly, the trained flow can be used within targeted free energy perturbation (TFEP)[44] to estimate the free energy difference between the base and target states, $\Delta f_{qp} = f_p - f_q$. Within the augmented flow framework, the partition function of the generated joint distribution depends on both physical and auxiliary variables. Since the target and base distributions of the auxiliary system are Gaussians, the partition function factorizes (see Supplementary Note 1) and the free energy of an augmented system is given by

$$f_\mu^{\mathrm{aug}} = f_\mu + f_\mu^{\mathrm{aux}}, \qquad (6)$$

where $f_\mu$ and $f_\mu^{\mathrm{aux}}$ denote the free energy of the physical and auxiliary system, respectively. The corresponding TFEP estimator for the augmented system is

$$\begin{aligned} \Delta f_{qp}^{\mathrm{aug}} = f_p^{\mathrm{aug}} - f_q^{\mathrm{aug}} &= f_p + f_p^{\mathrm{aux}} - (f_q + f_q^{\mathrm{aux}}) \\ &= \Delta f_{qp} + \Delta f_{qp}^{\mathrm{aux}}. \end{aligned} \qquad (7)$$

If $\eta_q = \eta_p$, as chosen in this work, then $\Delta f_{qp}^{\mathrm{aux}} = 0$ and $\Delta f_{qp}^{\mathrm{aug}} = \Delta f_{qp}$.

Flow-based approaches also allow to compute Gibbs free energies corresponding to the *NPT* ensemble, in which the shape of the simulation box, $\mathbf{h} \in \mathbb{R}^{3 \times 3}$ with $V = \det \mathbf{h}$, changes during the simulation at constant pressure $P$[1]. While prior work[17,18] studied flows modeling shape and configurational distributions simultaneously, in our current approach, the Gibbs free energy is computed via a Legendre transformation as $G = \min_{\mathbf{h}}[F(\mathbf{h}) + \det(\mathbf{h})P]$[45]. To obtain a reliable estimate for $F(\mathbf{h})$, we train the flow in a shape-conditional fashion[18,46], optimizing the flow to generate samples for varying box dimensions. The Gibbs free energies computed in this manner also enable the calculation of derived quantities, such as isothermal compressibilities (see Supplementary Notes 3 and 4 for details on the conditional training and derived quantities, respectively).

The principle advantages of our proposed BG architecture based on local augmented flows are reduced training times, improved sampling efficiencies, and, most importantly, its ability to generalize to larger systems. In the following, we demonstrate the applicability of our method across a range of crystal structures for various materials systems, exemplified by the Lennard-Jones (LJ) potential and multiple parameterizations of the Stillinger-Weber potential[28], including ice phases of monatomic water (mW) and silicon. Computational details are provided in "Methods".

## Size transferable training and evaluation

We illustrate the effectiveness of the size transferability by applying local BGs, trained on cubic mW ice with $N = 216$ particles and face-centred cubic (FCC) LJ with $N = 256$, to systems containing 512, 1000, and 1728 particles for cubic mW ice and 500, 864, 1000, and 1372 particles for FCC LJ.

Figure 2 shows the radial distribution functions (RDFs) and potential energy histograms for the largest investigated system sizes, as computed from samples generated by the BGs, MD, and drawn from the base distribution. For both systems, the RDFs produced by the BGs are nearly indistinguishable from those obtained via MD over length scales going far beyond the size of the training systems. While by

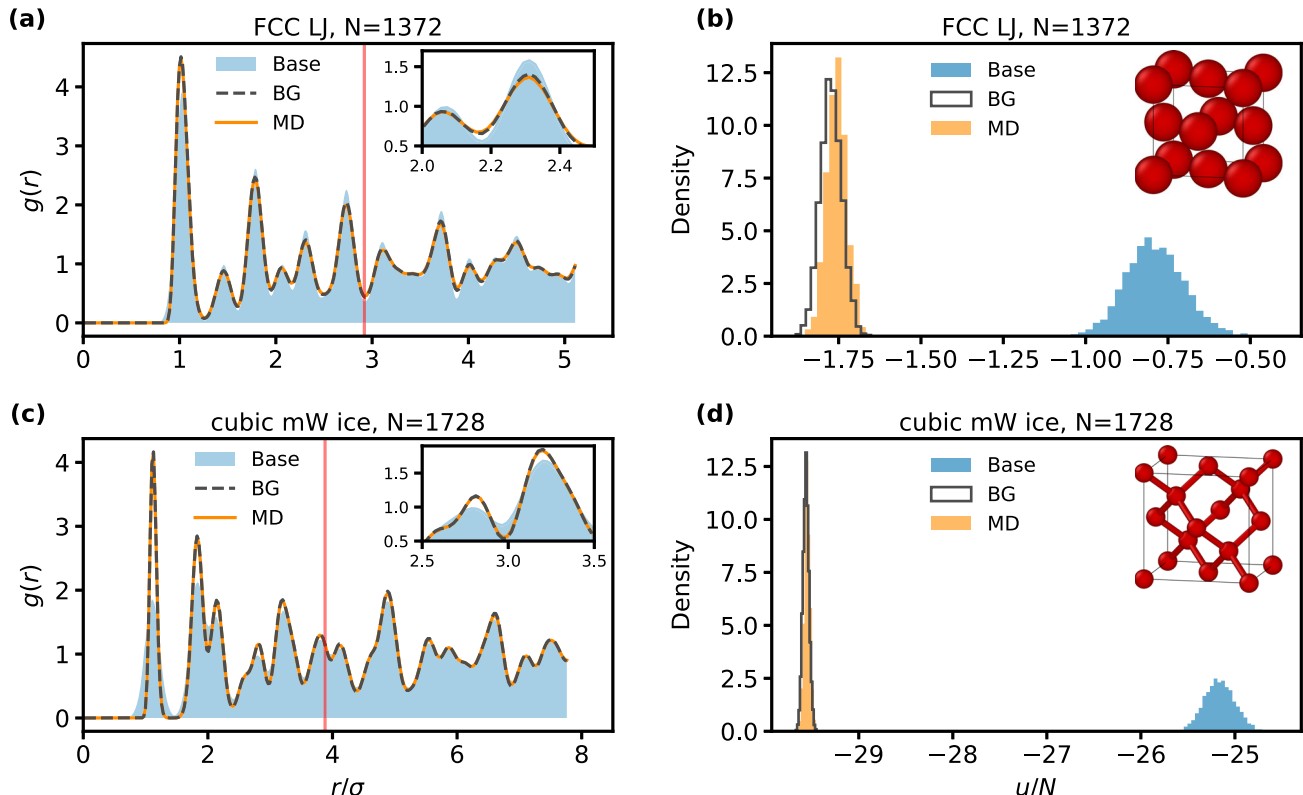

**Fig. 2 | Radial distribution functions and energy histograms. a, c** show the radial distribution functions (RDFs) in units of the interaction length $\sigma$ for FCC LJ with $N = 1372$ and cubic mW ice with $N = 1728$, respectively, as obtained from MD, the base distributions, and the local BGs. **b, d** show the corresponding reduced energy histograms. The local BG results were computed using models trained with $N = 216$ (mW) and $N = 256$ (LJ). The red line indicates half the box length of the training system, which is the largest distance for which the RDF can be evaluated for this system. For the transferred systems, the RDF is similarly evaluated up to their corresponding half-box length. No reweighting was applied. Source data are provided as a Source Data file.

design the RDF of the base distribution already captures the main features of the target, the corresponding configurations do not reflect the correct correlations between particle positions in the target potential, which is evident from the pronounced differences in the energy distributions of the base and MD ensembles. In contrast, the local BGs accurately reproduce both structural and energetic features in both systems, yielding excellent agreement in the histograms already without reweighting. These results clearly demonstrate that coordinate transformations based on local environments are entirely sufficient for generating accurate equilibrium structures and are reliably transferable to larger system sizes.

While structural and energetic accuracy are necessary conditions for accurate free energy estimates, they alone do not guarantee high sampling efficiency or precise free energy calculations. A more rigorous measure is the effective sample size (ESS) (see Methods) evaluated across the various system sizes within the transferable framework and summarized in the lower part of Table 1. As the system size increases, the ESS naturally decreases, as even constant per-particle errors in the extensive potential energy lead to exponentially amplified errors in the Boltzmann distribution. But even for large systems such as cubic mW ice with $N = 1728$ and FCC LJ with $N = 1372$, the ESS of the local BGs is large enough to ensure reliable and highly accurate statistics of ensemble averages.

Importantly, the ESS for large systems is significantly higher than that obtained with global approaches, while the computational cost is significantly reduced. For mW ice with $N = 512$, global BGs reported in refs. 15, 17 required more than 330 GPU days of training until convergence, yet achieved ESS values of only around 0.2%. In contrast, for the mW system with $N = 216$, our local BG reaches convergence in approximately four GPU days on the same hardware (see Supplementary Note 6 for details on training times), while yielding ESS values for the 512-particle system up to an order of magnitude higher than those obtained with the global BG. Moreover, scaling global architectures to larger systems becomes computationally prohibitive, making particle counts beyond 500 effectively impossible due to excessive training costs and deteriorating sampling efficiency.

The improved training performance of the local architecture compared to a global one is further illustrated in Fig. 3, evaluating the ESS of both types as a function of training time for three different systems (cubic mW ice with $N = 64$ and $N = 216$, and FCC LJ with $N = 216$). For the global BGs, the architecture proposed in Ref. 15 was used, which employs an attention-based coupling flow. Across all systems, the local BGs consistently achieve higher ESS than the global ones at any given training time. This is particularly evident for the larger system sizes, demonstrating the superior scaling of the local BGs with number of particles. The local BGs also exhibit substantially lower variance in ESS, indicating that reliable estimates can be obtained even with small sample sizes and remain stable as the number of samples increases.

These trends are corroborated by the ESS obtained from the converged models, which are reported in the upper part of Table 1. Remarkably, the marginal ESS of the physical system, which is the metric of primary interest, exceeds 80% for mW ice with $N = 64$ and over 40% for $N = 216$, and reaches about 7% for FCC LJ with $N = 256$, highlighting the significantly improved efficiency of our local flow architecture in capturing the relevant short-range environmental information. Where available, we have also provided the ESS of the global BGs as reported in Ref. 17, which were obtained using models

comprising twice as many layers and trained significantly longer. While the final ESS reached by these models is slightly higher than the ones we obtained for the global BGs within this work, they remain far lower than those of the local BGs.

We note that the BGs do not perform equally well across all systems. For example, both local and global BGs achieve significantly higher sampling efficiencies for the cubic mW ice compared to FCC LJ, despite only a modest increase in particle number. However, we also observed that certain crystal structures within the same potential can exhibit rather different sampling efficiencies, which asks for further investigation beyond the scope of this work.

### Free energy estimation

A key advantage of a trained BG is that it enables a direct evaluation of free energy differences within the framework of TFEP[44], as discussed in "Methods". Utilizing our local architecture, BGs trained on relatively small systems allow for an accurate evaluation of absolute Helmholtz and Gibbs free energies for far larger systems, which reduces the

**Table 1 | Effective sample size (ESS) in percent for local and global BGs for cubic mW ice and FCC LJ with different number of particles**

|  | Global | Local – joint | Local – marginal | Global[17] |
|---|---|---|---|---|
| *Training* |  |  |  |  |
| mW (*N*=64) | 45.9 ± 6.5 | 64.7 ± 0.7 | **82.7 ± 0.7** | 53.7 |
| mW (*N*=216) | 1.0 ± 0.9 | 16.4 ± 1.5 | **41.7 ± 1.6** | 6.8 |
| LJ (*N*=256) | 0.03 ± 0.03 | 2.0 ± 0.8 | **7.3 ± 2.6** | - |
| *Transfer (only local)* |  |  |  |  |
| mW (*N*=512) | - | 0.6 ± 0.4 | **1.8 ± 0.7** | 0.2 |
| mW (*N*=1000) | - | 0.04 ± 0.4 | **0.1 ± 0.1** | - |
| mW (*N*=1728) | - | 0.02 ± 0.02 | 0.02 ± 0.02 | - |
| LJ (*N*=500) | - | 0.1 ± 0.1 | **0.4 ± 0.3** | - |
| LJ (*N*=864) | - | 0.03 ± 0.02 | 0.03 ± 0.03 | - |
| LJ (*N*=1372) | - | 0.01 ± 0.01 | **0.02 ± 0.01** | - |

The top three rows show the results obtained for flow models of comparable size which were trained for 1M steps. Uncertainties are computed as the standard deviation over 4 different models each evaluated 5 times with 50k samples. The fourth column shows results reported in Ref. 17, which used twice as many layers and were trained significantly longer. The results of the local BGs in the bottom six rows were obtained using local models trained on systems with N = 216 for cubic mW ice and N = 256 for FCC LJ. **Bold** font indicates the best results.

computational effort dramatically compared to traditional free energy estimators such as MBAR[47].

The left panel of Fig. 4 shows the absolute free energy of the cubic ice phase as a function of the number of particles computed with the BG trained on 216 particles. In addition to the joint free energy estimates, we also present values computed from the marginal generated density and further compare to high-accuracy MBAR estimates obtained from MD simulations interpolating between the Einstein crystal and the physical system of interest[48]. Joint and marginal free energy estimates from the BG show excellent agreement with the reference values for system sizes up to 1000 particles, deviating by less than $10^{-3} k_B T$ per particle. For the largest system size ($N = 1728$), the joint estimates slightly overestimate the true values, while the marginal evaluation partially corrects for this, yielding more accurate results in agreement with the ESS reported in Tab. 1. In this context, it is worth mentioning that it is difficult to verify the reliability of a given free energy estimate at low ESS, since the variance of the estimates can remain very small even when ESS is low (see Supplementary Note 5). Based on the examples considered, a mean ESS of 0.1% or higher appears to be a sufficient criterion to ensure deviations in the free energy estimates remain below $10^{-3} k_B T$ per particle when using $5 \cdot 10^4$ samples. A similar behavior was observed for the FCC LJ crystal, with the corresponding free energy estimates reported in Supplementary Note 5.

Obtaining accurate free energy difference between different crystalline phases is particularly challenging but crucial to explore phase diagrams. To reach a converged value of the free energy difference between cubic and hexagonal mW ice, system sizes up to 1728 particles are necessary, as shown in the right panel of Fig. 4. Both joint and marginal estimates of $\Delta f$ are in very close agreement with the reference values, where the good performance of the joint estimates is likely due to error cancellation between the two phases. The large number of particles necessary to fully converge the free energy difference between the cubic and hexagonal phase underscores the importance of being able to evaluate extended system size which is infeasible with global architectures.

To quantify the computational savings associated with the size-transferable free energy estimation, we compare the number of energy evaluations required to the one of the standard MD+MBAR approach using typical parameters to interpolate between the Einstein crystal and the physical system[21,49,50]. The MBAR setup requires approximately 50 intermediate states with $10^3$ uncorrelated samples per state, obtained by storing one sample every $10^3$ MD steps, resulting in a total of around $5 \cdot 10^7$ energy evaluations. In addition, MBAR evaluates all $5 \cdot 10^4$ stored samples across all 50 intermediate potentials, resulting in additional

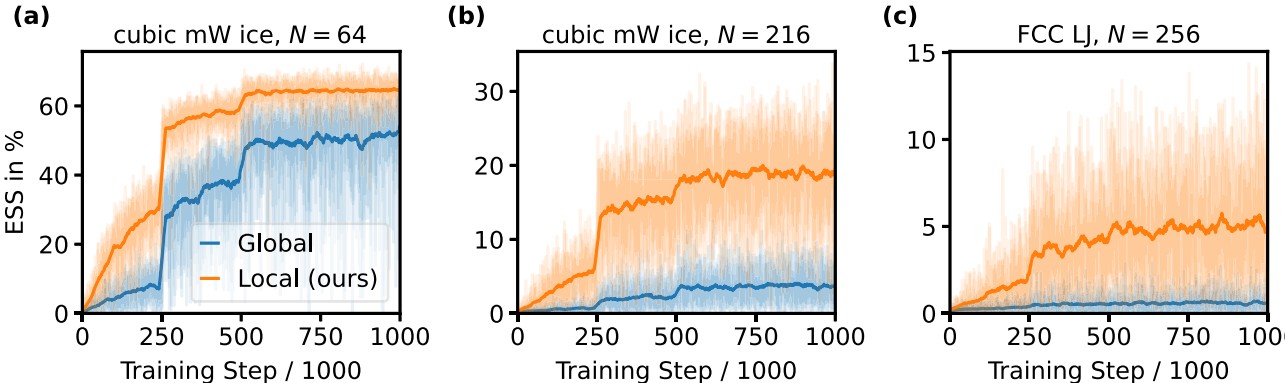

**Fig. 3 | Effective sample size.** Effective sample size (ESS) plotted against the number of training steps for global and local BGs of **a** cubic mW ice with *N* = 64 and **b** *N* = 216, and **c** FCC LJ with *N* = 256. The ESS was evaluated every 1k steps using 1k samples. Solid lines represent running averages of ESS across five independent training runs, with thin, light-colored lines indicating the raw ESS values from individual runs. For the local BGs, the joint efficiency is shown. Learning rate reductions were applied after 250k and 500k steps. Source data are provided as a Source Data file.

**(a)**

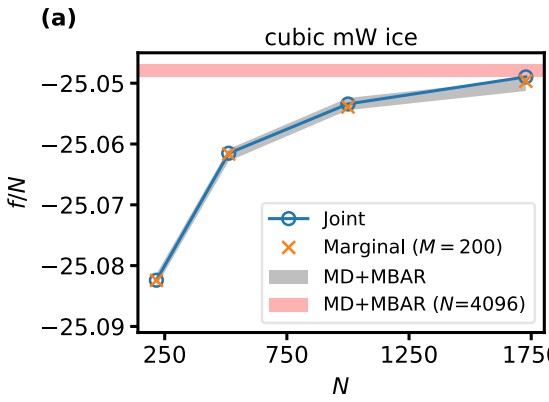

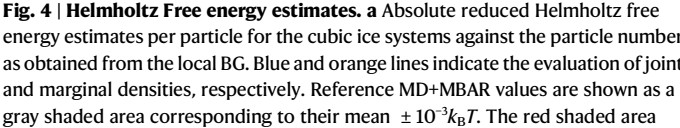

**(b)**

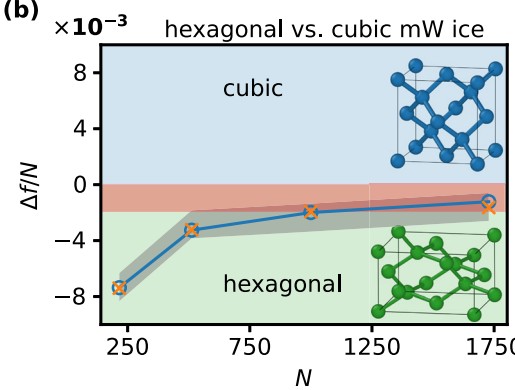

**Fig. 4 | Helmholtz Free energy estimates. a** Absolute reduced Helmholtz free energy estimates per particle for the cubic ice systems against the particle number as obtained from the local BG. Blue and orange lines indicate the evaluation of joint and marginal densities, respectively. Reference MD+MBAR values are shown as a gray shaded area corresponding to their mean $\pm 10^{-3} k_B T$. The red shaded area corresponds to the MD+MBAR results evaluated at $N = 4096$. BG results were obtained using a model trained at $N = 216$. Uncertainties are smaller than the marker size (see Supplementary Note 5). **b** Reduced free energy difference between cubic and hexagonal ice, $\Delta f = f_{hex} - f_{cubic}$, against the particle number. Shaded areas have the same meaning as for the left plot. Source data are provided as a Source Data file.

$2.5 \cdot 10^6$ energy evaluations. Training the flow model until convergence involves a comparable number of evaluations, approximately $6 \cdot 10^7$ for $5 \cdot 10^5$ training steps with batches of 128 samples (see Fig. 3). However, the training is performed for small system sizes only. Once trained, evaluating the free energy with the BG requires only a few tens of thousands of samples and the corresponding energy evaluations for the large system. In contrast, MD+MBAR necessitates several tens of millions of energy evaluations for each system size, with the computational cost scaling quadratically for pairwise potentials and even more steeply for quantum mechanical methods such as density functional theory and higher level electronic structure methods[51]. The computational savings of the flow-based approach are further amplified when performing convergence checks, which can be accomplished with a single model. Even further efficiency gains are possible by conditioning the model on external parameters, enabling reuse across multiple thermodynamic conditions[18], as exemplified below.

While BG models are highly efficient compared to MD+MBAR in terms of the number of required energy evaluations, this advantage does not directly translate to wall clock time for a single free energy estimate. For the cubic ice system studied here with $N = 1728$, a full MD +MBAR free energy estimate only takes a few hours, including simulations at all intermediate states and the MBAR evaluation, whereas training the flow model to convergence required approximately 8 GPU days (see Supplementary Note 6 for details). However, this comparison is specific to inexpensive interaction models such as mW, which have a cost per MD step on the order of $10^{-7}$ s per atom. Machine learning interatomic potentials, today the workhorse of computational materials science, are typically more expensive than empirical potentials by a factor of $10^3$ to $10^{4}$[52]. In these regimes, energy evaluations dominate the cost, while the overhead from optimization and neural network evaluation for the flow model remains largely independent of the potential. Consequently, the reduced number of energy evaluations required by the BG approach is expected to make the relative wall clock cost considerably more favorable, or even cheaper, than conventional MD+MBAR calculations.

### Including volume fluctuations
To illustrate the efficiency and accuracy of our approach in the isothermal-isobaric ensemble, we determine the Gibbs free energy difference between FCC and the hexagonal close-packed (HCP) LJ crystals at constant pressure. Similar to the hexagonal and cubic ice phases, these structures are known to have extremely small free energy differences, which require large system sizes to converge. Crucially, the relative stability of FCC and HCP depends on the cutoff

radius of the LJ potential and small cutoffs can even lead to qualitatively wrong predictions of the stable phase[22,53]. The local BGs for each phase are trained in a volume-conditional way (see Supplementary Note 3) using simulation cells comprising 180 particles. The trained BGs are subsequently applied to systems with $N = 1080$ to evaluate the Gibbs free energy over a range of cutoff radii.

Figure 5 summarizes the BG-based Gibbs free energy estimates for the two crystalline phases. The left panel shows the difference in reduced Gibbs free energy per particle between HCP and FCC as a function of the cutoff radius. The predictions of the BGs (based on joint density estimates) show excellent agreement with reference values obtained from MD combined with MBAR. Consistent with previous studies[22,53], a strong dependence of the free energy difference on the cutoff radius is revealed, leading to a qualitative change in the predicted stable phase. As the cutoff increases, this sensitivity decreases, emphasizing the need for large simulation cells to accommodate sufficiently large cutoff radii. Additionally, the particle densities obtained via the Legendre transformation using the BGs closely match the mean densities observed in *NPT* MD simulations, showing nearly identical dependence with respect to the cutoff radius (right panel of Fig. 5). Remarkably, the BG-based estimates yield highly accurate results over the entire range of cutoff radii. This strongly suggests that the local flow models can be trained using a single, short cutoff in a relatively small system, yet still be transferable to much larger systems while continuing to generate accurate configurations.

### Conditioning on atom type
As another important application, we show that the flow can be trained conditioned on parameters of the interaction potential, which allows a single model to generalize over a whole class of systems. As an example, we employ the Stillinger-Weber (SW) potential[28], combining a pair potential and a three-body term. The strength of the three-body interaction is controlled by a factor $\lambda_3$, yielding a total potential energy function

$$U_{SW}(\mathbf{x}) = \Phi_2(\mathbf{x}) + \lambda_3 \Phi_3(\mathbf{x}). \qquad (8)$$

For each system of interest, the SW potential is defined in terms of characteristic length and energy scales, which are tuned to reproduce specific properties of the system. However, when expressed in reduced units, the only effective parameter distinguishing different parameterizations is the strength of the three-body interaction, $\lambda_3$[54]. Importantly, different values of $\lambda_3$ stabilize or destabilize different crystal structures.

To study the phase diagram under varying three-body strengths and system sizes, we train local BGs for diamond cubic and $\beta$-tin structures with $N = 216$, and for the body-centered cubic (BCC) structure with $N = 128$. The BGs are trained conditioned on $\lambda_3 \in [15, 24]$ and the shape of the simulation box, allowing the model to generalize across the full class of SW potentials and cover an entire range of different materials, as well as to evaluate Gibbs free energies. Figure 6 shows the Gibbs free energies at zero pressure of the three different crystal structures as a function of $\lambda_3$ for the system sizes used during training (left panel) and transferred to $N = 1000$ for diamond cubic and $\beta$-tin and 686 for BCC (right panel). Across the entire range of $\lambda_3$, the free energy estimates of the BGs align very closely with reference values from MD+MBAR. While the transition point between the cubic and $\beta$-tin structures is largely independent of system size, a slight shift is observed in the transition point between the BCC and $\beta$-tin structures, moving towards lower values of the three-body interaction

strength for larger system sizes. To capture this subtle effect, it is imperative to be able to determine highly accurate free energies for rather large system sizes. An additional advantage of the flow-based approach is that the models can be trained across all structures over the full $\lambda_3$ spectrum, even though some structures are only stable in certain parts of this region in MD simulations.

## Silicon phase diagram

Focusing on the SW parametrization for silicon (Si), we train local BGs conditioned on both temperature and the shape of the simulation box to map out the $(T, P)$-phase diagram. Specifically, we are aiming to determine the coexistence line between the diamond cubic and $\beta$-tin phases. Both training and evaluation of the BGs were performed at $N = 216$. While size-transferability was again excellent for the cubic structure, the $\beta$-tin structure appeared to be more challenging already at the training size, limiting the accuracy of free energy estimates in

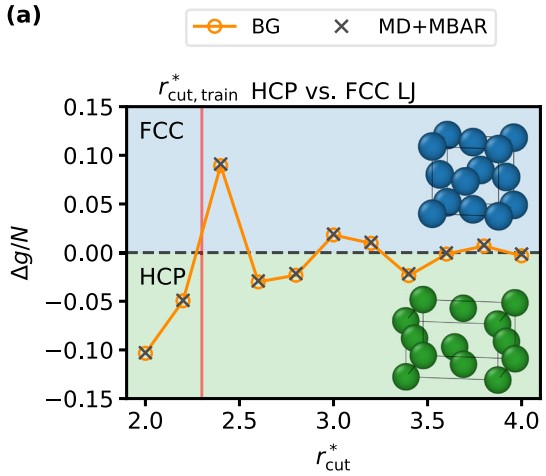

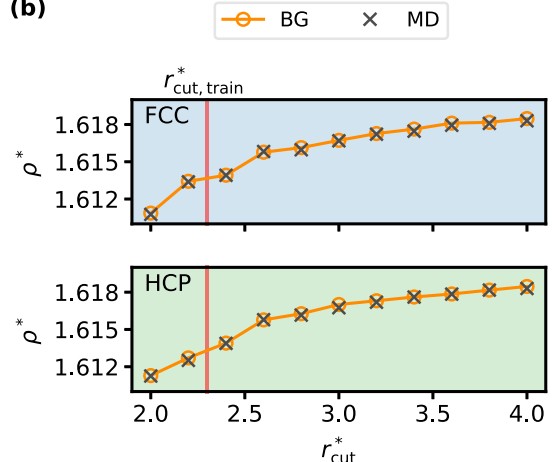

**Fig. 5 | Gibbs free energy estimates. a** Reduced Gibbs free energy difference $\Delta g = g_{HCP} - g_{FCC}$ per particle between HCP and FCC crystal structures with 1080 particles in the LJ potential as a function of the cutoff radius. Shown are BG-based predictions (joint density estimates) alongside reference results from MD+MBAR. The red vertical line denotes the cutoff radius applied during training with 180 particles (quantities marked with an asterisk are expressed in LJ units, see

Supplementary Note 2). **b** Densities $\rho^*$ as obtained from the BGs (circles) and mean densities from MD *NPT* simulations (crosses) for FCC (upper panel) and HCP (lower panel). Error bars for both BG and MD+MBAR, estimated from three independent runs each, are smaller than the plotted marker size. Source data are provided as a Source Data file.

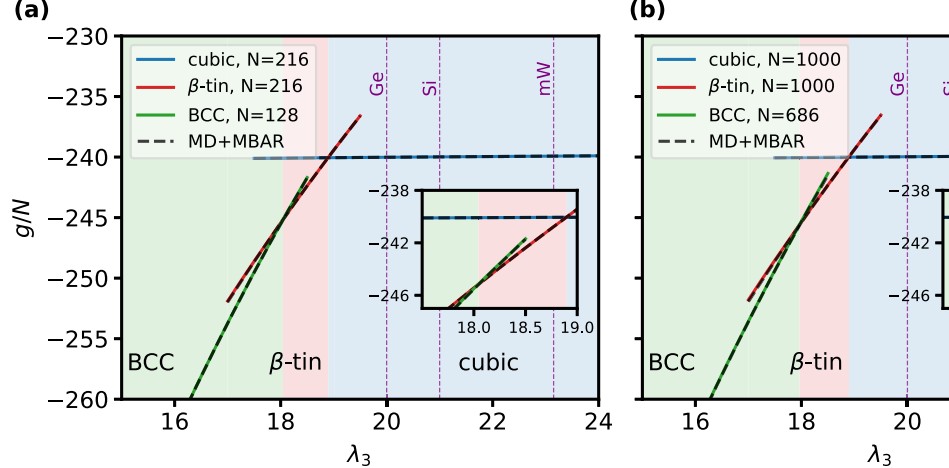

**Fig. 6 | Phase diagram SW potential. a, b** show the Gibbs free energies per particle for small and large particle numbers, respectively, as obtained from the local BGs at zero pressure for different crystal structures as a function of the three-body interaction strength $\lambda_3$. Vertical dashed lines indicate the value of this parameter in the parametrizations for germanium, silicon, and monatomic water. The color of

the shaded areas corresponds to the most stable structure. MD+MBAR values are shown as black dashed lines. Error bars for both BG and MD+MBAR, estimated from three independent runs each, are smaller than the plotted line width. Source data are provided as a Source Data file.

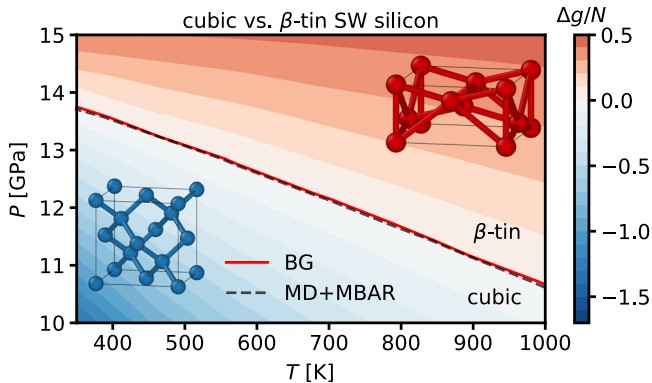

**Fig. 7 | Silicon phase diagram.** Si phase diagram for $N = 216$ particles, computed using local BGs and MD+MBAR. The color scale represents the reduced Gibbs free energy difference per particle between $\beta$-tin and diamond cubic phases, $\Delta g = g_{cubic} - g_{\beta\text{-tin}}$, as obtained from the local BGs. The red line marks the coexistence line between the two phases, while the black dashed line indicates the corresponding result from MD+MBAR estimates. Error bars for both BG and MD+MBAR, estimated from three independent runs each, are smaller than the plotted line width. Source data are provided as a Source Data file.

much larger systems. To improve the accuracy of the predicted phase diagram, the Gibbs free energies of the $\beta$-tin phase were evaluated based on the marginal density using $M = 200$.

Even without transferring to larger systems, the conditional training provides substantial cost amortization across thermodynamic states, since reference MD+MBAR calculations require fully converged simulations on at least an $7 \times 7$ grid to cover the full $(T, P)$-range, whereas the local BGs enable efficient evaluation of Gibbs free energies across all thermodynamic states. The resulting phase diagram of Si is presented in Fig. 7. It captures the competition between the cubic and $\beta$-tin phases across a wide range of conditions and accurately predicts the corresponding coexistence line. Validation against free energy estimates obtained from MD+MBAR shows excellent agreement, as reflected in the practical indistinguishability of the coexistence lines.

## Discussion

We introduced a local flow architecture for training generative models that enables equilibrium sampling of materials systems, scaling seamlessly from small training cells to significantly larger systems exceeding 1000 particles. Our approach surpasses previous global architectures by achieving faster training times as well as higher sampling efficiencies. Crucially, the flow models exhibit transferability across system sizes and allow to effectively sample both short- and long-range interaction potentials, yielding highly accurate absolute free energy estimates across a variety of materials systems. By conditioning on thermodynamic variables and interaction parameters, the training process is further amortized, enabling direct and efficient evaluation of phase stability across temperature, pressure, and composition. The presented results highlight the potential of our architecture to serve as a powerful and scalable tool for studying complex materials across a wide range of thermodynamic conditions.

Future directions include extending this approach to efficiently sample liquid phases and developing models with broad generalization across chemical space. In this context, it will be important to investigate why the models exhibit different performances for different combinations of potentials and structures, to better understand and guide the design of generative approaches. A particularly interesting extension beyond the present formulation for atomic solids would be the development of models applicable to molecular crystals. While conceptually conceivable within the developed approach, this would require more elaborate architectural modifications to account

for orientational degrees of freedom, such as those arising from rigid body constraints. Another important direction for future work is the development of models capable of capturing long-range interactions, such as those arising from electrostatics, which cannot be represented within any purely local formulation. Together, these developments could significantly expand the scope and impact of generative modeling in materials science.

## Methods
### The canonical ensemble

Within the canonical ($NVT$) ensemble, with fixed number of particles $N$, temperature $T$, and volume $V$, the equilibrium distribution is defined by the Boltzmann distribution[1,2]

$$p(\mathbf{x}) = \frac{e^{-u_p(\mathbf{x})}}{Z_p} , \qquad (9)$$

where $u_p(\mathbf{x}) = \beta_p U_p(\mathbf{x})$ is the reduced potential with the potential energy $U_p(\mathbf{x})$ of configuration $\mathbf{x} \in \mathbb{R}^{3N}$, the inverse temperature $\beta_p = 1/k_B T_p$, and Boltzmann's constant $k_B$. The normalization constant $Z_p = \int e^{-u_p(\mathbf{x})} d\mathbf{x}$ is known as the configurational partition function, from which the reduced Helmholtz free energy $f_p = \beta_p F_p$ is defined as

$$f_p = - \log Z_p . \qquad (10)$$

The free energy further includes a momentum contribution, which can be evaluated analytically[1], and all absolute free energies reported in the current study include the momentum contribution. While this work focuses on equilibrium sampling in the canonical ensemble, other ensembles can be modeled by adjusting the reduced potential[3,47].

### Boltzmann generators

Boltzmann Generators[3] use a flow transformation[4,5,32,55] $\mathcal{F}_\theta : \mathbb{R}^{3N} \to \mathbb{R}^{3N}$ to transform samples $\mathbf{x}$ from a base distribution $q(\mathbf{x})$ into samples $\mathbf{x}' = \mathcal{F}_\theta(\mathbf{x})$ of a distribution $q_\theta(\mathbf{x})$ that approximates $p(\mathbf{x})$. The base distribution is chosen so as to permit simple generation of random variables from it, for example a multivariate normal distribution. The exact-likelihood property of normalizing flows allows to evaluate the density of generated samples via the change-of-variable theorem

$$q_\theta(\mathbf{x}') = q(\mathbf{x}) | \det J_{\mathcal{F}_\theta}(\mathbf{x})|^{-1} , \qquad (11)$$

such that the generated samples can be reweighted to the target distribution. The mapping $\mathcal{F}_\theta$ is commonly implemented using continuous-time formulations such as neural ordinary differential equations (ODEs)[32,56], or in a discrete fashion based on coupling flows[38,39]. While continuous flows offer smooth and expressive mappings by modelling transformations as solutions to ODEs, the evaluation of the Jacobian determinant, $\det J_{\mathcal{F}_\theta}$, often involves high computational costs, especially in high dimensions. Coupling flows achieve efficient and exact likelihood evaluation, but may require deep architectures to achieve high expressiveness.

In contrast to continuous flows, an important advantage of coupling flows is their compatibility with likelihood-based optimization schemes, allowing the flow to be trained, requiring only knowledge of the potential energy function of the target distribution but no i.i.d. samples. Specifically, the flow is trained by minimizing the KL divergence between generated and target distribution, $D_{KL}(q_\theta \| p)$. For the equilibrium distribution of the canonical ensemble (Eq. (9)), the corresponding loss function reduces to[3]

$$\mathcal{L}_{qp}(\theta) = - \mathbb{E}_{\mathbf{x} \sim q}\left[\log w(\mathbf{x})\right] \geq \Delta f_{qp} , \qquad (12)$$

where $\Delta f_{qp}$ is the free energy difference between $q$ and $p$. The importance weights $w$ are defined using the reduced potentials of the

base and target, $u_q$ and $u_p$, as

$$w(\mathbf{x}) = e^{u_q(\mathbf{x}) - u_p(\mathcal{F}_\theta(\mathbf{x})) + \log|\det J_{\mathcal{F}_\theta}(\mathbf{x})|}, \qquad (13)$$

such that $w(\mathbf{x}) \propto p(\mathcal{F}_\theta(\mathbf{x}))/q_\theta(\mathcal{F}_\theta(\mathbf{x}))$. Since $\Delta f_{qp}$ is typically unknown during training, $\mathcal{L}_{qp}$ offers only limited insight into convergence. An additional useful diagnostic is the Kish effective sample size (ESS)[57]

$$\text{ESS} = \frac{\left[\sum_i w(\mathbf{x}_i)\right]^2}{\sum_i \left[w(\mathbf{x}_i)\right]^2}. \qquad (14)$$

The ESS provides an approximate measure of how many uncorrelated samples would yield a Monte Carlo estimator of comparable statistical quality, making it a problem-independent indicator of the flow model's sampling performance.

### Targeted free energy perturbation

In the context of targeted free energy perturbation[44], the free energy difference between base and target, $\Delta f_{qp}$, can be directly obtained from the trained BG via

$$\Delta f_{qp} = -\log \mathbb{E}_{\mathbf{x} \sim q}[w(\mathbf{x})]. \qquad (15)$$

BGs can thus offer significant improvements in computational efficiency over traditional free energy estimators, such as free energy perturbation (FEP)[58] and multistate Bennett acceptance ratio (MBAR)[47], which require a chain of intermediate distributions between $q$ and $p$ to ensure sufficient overlap for convergence. In this context, BGs can be seen as a practical implementation of TFEP[44], analogous to the learned free energy perturbation approach introduced in Ref. 14.

### Computational settings

In the *NVT* ensemble, cubic and hexagonal mW ice systems were modeled at $T = 200K$ at a particle density of $\rho = 0.0336\text{Å}^{-3}$. The FCC LJ crystal was modeled at a reduced temperature of $T^* = 2.0$ and a reduced density of $\rho^* = 1.28$ (see Supplementary Note 2 for details on reduced units).

In the *NPT* ensemble, the FCC and HCP LJ crystals were simulated at a reduced temperature of $T^* = 0.2$ and the Gibbs free energy was evaluated at $P^* = 150$. The phase diagram of the SW potential was simulated at a reduced temperature of $T^* = 0.01$.

In the *NVT* ensemble, four independent models were trained per system and yielded identical performance with very small statistical uncertainties (see Supplementary Note 5). Consequently, we trained only one model per system to obtain the results of the *NPT* ensemble, with error bars estimated from three independent evaluations of this single model.

### Data availability

The molecular dynamics data generated in this study were used solely to compute reference free energies for comparison with the model predictions. We have not generated any dataset for model training. The raw simulation data are not stored or deposited, as the underlying calculations use inexpensive potentials and can be efficiently recomputed if required. The source code used for the MD simulations is openly available in Zenodo at https://zenodo.org/records/19349064[59]. Source data of the results shown in the figures are provided with this paper.

### Code availability

The flow models developed in this work are available at https://github.com/maxschebek/bgmat[60]. Models are built and trained using JAX[61], Haiku[62] and Distrax[63]. All MD simulations were performed using the OpenMM package[64].

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

## Acknowledgements

MS thanks the ICC at the Flatiron Institute for hospitality while part of this research was carried out there. The computations reported in this paper were in part performed using resources made available by the Flatiron Institute. The Flatiron Institute is a division of the Simons Foundation. We

thank Leon Klein, Michael Plainer, and Emil Hoffmann for insightful discussions.

## Author contributions

M.S.: Conceptualization (equal); Formal Analysis (equal); Investigation (lead); Software (lead); Visualization (lead); Writing—original draft (lead); Writing—review and editing (equal). F.N.: Conceptualization (equal); Project administration (equal); Writing—original draft (supporting); Writing—review and editing (equal). J.R.: Conceptualization (equal); Project administration (equal); Supervision (equal); Writing—original draft (equal); Writing—review and editing (equal).

## Funding

MS and JR acknowledge financial support from Deutsche Forschungsgemeinschaft (DFG) through grant CRC 1114 "Scaling Cascades in Complex Systems", Project Number 235221301, Project B08 "Multiscale Boltzmann Generators". JR acknowledges financial support from DFG through the Heisenberg Programme project 428315600. Open Access funding enabled and organized by Projekt DEAL.

## Competing interests

The authors declare no competing interests.
