## [Transparent Peer Review file · Nature Communications]

Scalable Boltzmann Generators for equilibrium sampling of large-scale materials

Corresponding Author: Mr Maximilian Schebek

Version 0:

Reviewer comments:

Reviewer #1

(Remarks to the Author)

Overall, the paper presents an architectural advance for Boltzmann sampling and makes a compelling case for size transferability in crystalline materials. However, a key validation is currently missing. In particular, the proposed architecture is only tested on atomic solids, while its claimed generality and practical relevance would critically require validation on molecular crystals, which involve orientational degrees of freedom, rigid-body constraints, and more complex local environments. Demonstrating either successful application to molecular crystals or clearly identifying the limitations of the approach in this setting is essential for assessing the true scope and robustness of the method, and should be addressed before the paper can be considered for acceptance. The revisions outlined below are also strongly recommended to strengthen the manuscript.

Major Revision Remarks

- 1 Equations (5) and (6), which decompose the augmented free energy into physical and auxiliary contributions, introduce many new symbols (e.g. Δf_{qp}) in a very compact form. Currently, most explanations are postponed to the SI. Given that these equations underpin the free energy calculations throughout the paper, the authors should provide clearer explanations and definitions directly in the main text rather than relegating them to supplementary material.
- 2 While the proposed architecture convincingly generalizes across system sizes, it remains unclear how well it generalizes across thermodynamic state points, particularly density. For example, can a model trained at one density be reliably applied at a different density? This question is crucial, especially since the paper relies on Legendre transforms to obtain Gibbs free energies, which implicitly require accurate Helmholtz free energies across a range of densities. A discussion or numerical test addressing this issue would substantially strengthen the paper.
- 3 The comparison between BGs and MD/MBAR focuses primarily on the number of energy evaluations. While informative, this metric alone is incomplete. BGs rely on expensive neural network evaluations, whereas classical potentials such as Lennard–Jones or mW are extremely cheap. A comparison in terms of actual wall-clock time or GPU hours would provide a more realistic assessment of the computational trade-offs involved.
- 4 The paper computes Gibbs free energies using a Legendre transform from Helmholtz free energies. To validate that this procedure is sufficiently accurate, I suggest computing thermodynamic response functions such as the isothermal compressibility from the Gibbs free energy and comparing them with reference values from MD. This would provide a stringent and physically meaningful test of the proposed framework.

Minor Revision Remarks

- 1 In the target distribution, the auxiliary variables (a) are modeled as Gaussian-distributed around the physical coordinates (x), i.e. $p(a|x) = \mathcal{N}(a; x, \eta^2 I)$. While this construction is standard in augmented flow frameworks, it would be good to explain here why this particular design is chosen and what intuition it provides.
- 2 The marginal distribution over physical coordinates is obtained by integrating out the auxiliary variables. Currently, the derivation is deferred almost entirely to the Supplementary Information. Given its central importance, the authors should explicitly outline the key steps of the integration.

3 η_q in $q(x) = \mathcal{N}(x; 0, \eta_q^2 I)$ is not defined.

(Remarks on code availability)

Reviewer #2

(Remarks to the Author)

This manuscript presents a substantial and timely advance in the development of Boltzmann Generators for the equilibrium sampling of condensed-phase materials. By introducing a local, size-transferable augmented flow architecture based on graph neural networks, the authors convincingly overcome one of the main limitations of prior BG approaches: their poor scalability and prohibitive training costs for large systems. The paper is technically sound, carefully executed, and, in my view, represents a significant contribution to the field of machine-learning-based molecular simulation.

Formulating the generative model in terms of local environments while retaining exact likelihoods through augmented coupling flows addresses a significant bottleneck in the current literature. The resulting architecture demonstrates transferability across system sizes, enabling models trained on relatively small cells to be applied to systems with over a thousand atoms. The numerical evidence supporting this claim is compelling. Structural observables, energy distributions, effective sample sizes, and free-energy estimates are validated against high-quality MD+MBAR reference calculations, showing excellent agreement across multiple model systems, including Lennard-Jones crystals, monatomic water ice, and Stillinger–Weber solids.

The demonstration that accurate absolute and relative free energies can be obtained for large crystalline systems at a computational cost competitive with, and in some cases superior to, state-of-the-art free-energy methods is particularly noteworthy. The discussion of size convergence and finite-size effects is thorough and convincing. Application to phase stability problems, such as FCC–HCP Lennard-Jones, ice polymorphs, and the silicon phase diagram, underscores the practical relevance of the approach. Conditioning on thermodynamic variables and interaction parameters further strengthens the study by illustrating how training costs can be amortized across state points and materials classes.

The manuscript is generally clear and well structured, and the figures are informative and thoughtfully designed. Any remaining weaknesses are minor.

For example, a slightly more explicit discussion of the limits of the locality assumption and the observed variability in sampling efficiency across different crystal structures could help guide future applications, where the number of degrees of freedom might scale with both chemical complexity and system size. Similarly, clarifying under which conditions accurate free-energy estimates remain possible despite very small effective sample sizes would benefit readers less familiar with targeted free-energy perturbation. These points are primarily matters of exposition and do not detract from the strength of the results.

In summary, this is a high-quality, impactful study that extends the practical applicability of Boltzmann Generators well beyond previous work. I strongly recommend publication in its present form, with only minor optional clarifications at the authors' discretion.

(Remarks on code availability)

Reviewer #3

(Remarks to the Author)

The present work introduces an architecture of normalizing flows (NFs) that are transferable in size for the purpose of studying crystals. The proposed architecture resembles the previous proposition of Ref [39]. However, here the authors restrict to a local neighborhood of each particle in the update rule at each layer. The authors show that this locality, implemented with a graph neural network, allows to train the model for a certain crystal size and then use it to sample a larger crystal, typically 5 times larger in their experiments. Training can be performed by optimizing the reverse KL in this application where the goal is to model the distribution close-by the perfect crystal and no multimodality is expected.

Despite using an augmented state space, that precludes exact likelihood computation of the model on the physical space, the authors describe means of evaluation of free energy differences through targeted free energy perturbation and report results that appear stable either when the marginal likelihood is estimated through a Monte Carlo or when relying on the free energy difference on the augmented space which should be equal to the marginal one (given the chosen joint law). Results are reported for cubic and hexagonal ice, FCC and HCP Lennard Jones and the Stillinger-Weber tunable potential with crystal sizes up to the order of 1000 particles. The NVT and the NPT ensembles can both be studied, in the latter case by training NFs conditional on the shape of the crystal cell. Experiments are sound and thorough, discussing various aspects of the method.

As expected the efficiency of sampling degrades as the size of crystals increases dropping to 0.02% for ~1500 particle crystals. This typical size appears to be just enough to converge the free energy estimates. Variability of the results for different crystals calls for more systematic assessment as it seems plausible that the method will fail in some cases (requiring larger cells). However, the comparison in terms of number of calls to the force compared to MD+MBAR is interesting, showing that the proposed method is not drastically cheaper when taking into account the training cost, but that

once training is done the free energy computations are cheaper in the considered examples (reaching ESS of 0.02% again) and all the more so if amortizing the training across some thermodynamic parameter.

The paper is overall well written, it describes a new methodology which is demonstrated in various experiments. The limits are honestly discussed in the text if not in the introduction and conclusion. I am in favor of accepting the paper while I note that further research will be needed to fully understand the potential of the method given the differences of performance observed for different crystals.

A few clarifications would still further improve the manuscript:

1- As far as I could follow the motivation for using augmented architectures in Ref [39] was to define an NF with invariant distribution with respect to some symmetries of the space. Here, the authors do not seem to be imposing any invariance (beyond the permutation invariance implied by the GNN). Could the authors clarify was they resort to augmented flows here? Making the following sentence more explicit may help "As coupling flows need to split coordinates either across particles or across spatial dimensions or a combination thereof, the calculation of full three-dimensional environments is prohibited [38]". Have they compared with the performance of traditional coupling flows?

2- The definition of Z_{aux} should be formally stated as it is a matter of convention. The most natural way is to define it as the partition of $a|x$ given the weight (S1) in SI. This yields $Z_{aux}(x)$, function of x , which indeed turns out to be independent of x consistently with the authors description.

3- In Figure 2, for the histogram of potentials for FCC LJ, the x axis could be better adapted so the agreement between BG and MD could be more finely assessed. There is a strange zoom out leaving significant blank space hurting the readability of the plot.

4- "For both systems, the RDFs produced by the BGs are nearly indistinguishable from those obtained via MD over length scales going far beyond the size of the training systems." p5 sec 3.1, I must be misunderstanding something, the plot seems to report $g(r)$ for up to L_{train} , meaning "up to" the size of the training system and not "far beyond"?

5- For figure 6: why the BCCs have lower numbers of particles than other phases?

(Remarks on code availability)

I have not run the code, but it features a readme and a tutorial notebook, functions appear to also be minimally commented.

Reviewer #4

(Remarks to the Author)

This paper addresses the scalability of Boltzmann generators. The method presented here allows for sampling of systems with more than a thousand atoms. Applied examples demonstrate the effectiveness of this approach.

The paper is well-written and it deserves to be published in Nature. Comm.

Minor points:

- Introduction: The authors might want to avoid defining their own work as "seminal."

=The systems presented here are far smaller than most systems simulated by MD/MC, such as biological systems. What are the prospects for sampling the latter using this method?

-Some sentences seem redundant and could be removed without altering the scientific content. For example, "The presented results highlight the potential of our architecture to serve as a powerful and scalable tool for studying complex materials across a wide range of thermodynamic conditions."

(Remarks on code availability)

Version 1:

Reviewer comments:

Reviewer #1

(Remarks to the Author)

The authors have addressed most of my comments. However, my main concern remains: the method has only been

demonstrated on atomic solids, while most real-world applications involve molecular crystals. I understand that a full extension is substantial work, but I still consider at least one example—e.g., SPC/E water—necessary to support your claims. Without such a demonstration, I cannot recommend publication.

(Remarks on code availability)

Reviewer #3

(Remarks to the Author)

I thank the authors for their detailed answer to my comments. I have no further concerns.

(Remarks on code availability)

Manuscript reference: NCOMMS-25-100979-T

Title: Scalable Boltzmann Generators for equilibrium sampling of large-scale materials

Author(s): Maximilian Schebek, Frank No e, and Jutta Rogal

-----Reply to the report of the first reviewer -----

We thank the reviewer for the thoughtful assessment of our manuscript. In order to address the points raised, we paste and quote the referee report and insert our reply.

Overall, the paper presents an architectural advance for Boltzmann sampling and makes a compelling case for size transferability in crystalline materials. However, a key validation is currently missing. In particular, the proposed architecture is only tested on atomic solids, while its claimed generality and practical relevance would critically require validation on molecular crystals, which involve orientational degrees of freedom, rigid-body constraints, and more complex local environments. Demonstrating either successful application to molecular crystals or clearly identifying the limitations of the approach in this setting is essential for assessing the true scope and robustness of the method, and should be addressed before the paper can be considered for acceptance.

We thank the reviewer for highlighting the importance of molecular crystals and fully agree that they represent a highly relevant and challenging class of materials for which scalable generative architectures are ultimately desirable. However, extending methods from atomic solids to molecular crystals is not a straightforward modification, but a substantial methodological step. The additional rotational, conformational, and intermolecular degrees of freedom in molecular crystals necessitate specialized representations and sampling strategies. Specifically, approaches tailored to molecular crystals (e.g., Refs. [S1, S2]) have required dedicated architectural developments and have resulted in studies entirely focused on rigid-body and internal coordinate formulations. A similar methodological distinction between atomic and molecular crystals is observed in the broader field of generative materials discovery. Recent advances (e.g., Refs. [S3, S4]) have been developed and validated primarily for atomic periodic systems. Extensions to molecular crystals have required additional methodological innovations, including recent work applying flow matching to molecular crystal generation [S5], which likewise constitutes a distinct line of development rather than a direct extension of models for atomic solids.

In this context, a complete realization of our framework for molecular crystals would constitute a substantial and self-contained research effort rather than a straightforward extension of the present work.

To nevertheless demonstrate conceptual compatibility, we outline at a high level how our approach could be integrated with the rigid-body flow architecture from Ref. [S1], without claiming a full implementation or validation.

Architecture review. A molecular crystal is modelled as a system of N rigid bodies, each composed of K beads with Cartesian coordinates $\mathbf{x} = (\mathbf{x}_0, \dots, \mathbf{x}_{K-1}) \in \mathbb{R}^{K \times 3}$. As a first step, the Cartesian coordinates of each rigid body are decomposed via a rigid-body transform $T(\mathbf{x}) = (\mathbf{x}_0, R, \Psi)$, where $\mathbf{x}_0 \in \mathbb{R}^3$ denotes the global translation, $R \in \text{SO}(3)$ the global rotation, and $\Psi \in \mathbb{R}^{3K-6}$ the $3K - 6$ internal degrees of freedom. The rotation R is then lifted to one of the two representing unit quaternions $\pm \mathbf{q} \in \mathcal{S}^3$.

A sign-flip equivariant coupling flow F acts on translation and quaternion,

$$F(\mathbf{x}_0, \mathbf{q}) = (\mathbf{x}'_0, \mathbf{q}'). \quad (\text{R1})$$

Using an arbitrary diffeomorphism $\xi(\cdot; \theta) : \mathbb{R}^3 \rightarrow \mathbb{R}^3$ and a diffeomorphism $\Phi(\cdot; \theta) : \mathcal{S}^3 \rightarrow \mathcal{S}^3$ which is

equivariant to quaternion sign flips, position and rotation updates are defined according to

$$\mathbf{x}'_0 = \xi(\mathbf{x}_0; \theta_\xi(\mathbf{q})), \quad (\text{R2a})$$

$$\mathbf{q}' = \Phi(\mathbf{q}; \theta_\Phi(\mathbf{x}'_0)), \quad (\text{R2b})$$

subject to the equivariance condition $F(\mathbf{x}_0, -\mathbf{q}) = (\mathbf{x}'_0, -\mathbf{q}')$. Here, θ_α denote conditioning networks. Projecting back to SO(3) yields $R' = g(\mathbf{q}')$, and the transformed Cartesian coordinates are reconstructed as $\mathbf{x}' = T^{-1}(\mathbf{x}'_0, R', \Psi)$.

Extension to the augmented setting. In the augmented setting, each rigid body \mathbf{x} may be augmented by auxiliary variables \mathbf{x}^a , which admit a corresponding representation $(\mathbf{x}_0^a, R^a, \Psi^a)$, where R^a is mapped to a unit quaternion \mathbf{q}^a . A possible adjustment of the coupling flow update in Eq. (R2) is then given by

$$\mathbf{x}'_0 = \xi_x(\mathbf{x}_0; \theta_\xi^x(\mathbf{x}_0^a, \mathbf{q}^a)), \quad (\text{R3a})$$

$$\mathbf{q}' = \xi_q(\mathbf{q}; \theta_\xi^q(\mathbf{x}_0^a, \mathbf{q}^a)), \quad (\text{R3b})$$

$$\mathbf{x}_0^{a'} = \Phi_x(\mathbf{x}_0^a; \theta_\Phi^x(\mathbf{x}'_0, \mathbf{q}')), \quad (\text{R3c})$$

$$\mathbf{q}^{a'} = \Phi_q(\mathbf{q}^a; \theta_\Phi^q(\mathbf{x}'_0, \mathbf{q}')). \quad (\text{R3d})$$

While the careful design of the conditioning networks represents a nontrivial challenge, the augmented formulation provides a clear conceptual advantage. In contrast to Eqs. (R2), the formulation in Eqs. (R3) allows the computation of interparticle distances, since all conditioning networks have access to both global translations and rotations. We have added the following to the discussion to emphasize the limitations of the current approach with respect to its applicability to molecular crystals:

“A particularly interesting extension beyond the present formulation for atomic solids would be the development of models applicable to molecular crystals. While conceptually conceivable within the developed approach, this would require more elaborate architectural modifications to account for orientational degrees of freedom, such as those arising from rigid body constraints.”

The revisions outlined below are also strongly recommended to strengthen the manuscript. Major Revision Remarks:

1. Equations (5) and (6), which decompose the augmented free energy into physical and auxiliary contributions, introduce many new symbols (e.g. Δf_{pq}) in a very compact form. Currently, most explanations are postponed to the SI. Given that these equations underpin the free energy calculations throughout the paper, the authors should provide clearer explanations and definitions directly in the main text rather than relegating them to supplementary material.

We thank the reviewer for pointing this out. In response, we have expanded the discussion in the main text to provide a clearer explanation of Eqs. (5) and (6) (in the revised manuscript Eqs. (6) and (7)). In particular, we now introduce the notation for the free energies and the free energy differences associated with the different systems more explicitly to make the decomposition of the augmented free energy and its role in the subsequent calculations more transparent:

“As detailed in Methods, the calculation of the (reduced) Helmholtz free energy associated to a distribution μ , $f_\mu = F_\mu/k_B T_\mu = -\log Z_\mu$, requires an estimate of the partition function Z_μ . While this quantity cannot be computed directly, the trained flow can be used within targeted free energy perturbation (TFEP) [42] to estimate the free energy difference between the base and target states, $\Delta f_{pq} = f_p - f_q$.”

2. While the proposed architecture convincingly generalizes across system sizes, it remains unclear how well it generalizes across thermodynamic state points, particularly density. For example, can a model trained at one density be reliably applied at a different density? This question is crucial, especially since the paper relies on Legendre transforms to obtain Gibbs free energies, which implicitly require accurate Helmholtz free energies across a range of densities. A discussion or numerical test addressing this issue would substantially strengthen the paper.

We thank the reviewer for raising this important point. The models used to compute the Legendre transform were not trained on a single density but *conditioned* on the simulation box \mathbf{h} (varying both shape and volume) and temperature, which enables them to generalize across densities and temperatures. Since these (T, \mathbf{h}) -conditioned models were used to compute the silicon phase diagram, which is in excellent agreement with the MD+MBAR reference results, there is evidence that they generalize accurately over the relevant thermodynamic states.

To demonstrate the generalization capability over densities more explicitly, we used the trained model of the cubic Si phase and evaluated its performance for free-energy estimation across a range of densities corresponding to the thermodynamic states considered in the phase diagram. We find that the model yields highly accurate Helmholtz free-energy estimates throughout this range. The corresponding analysis and plot are provided in Supplementary Note C of the revised SI.

3. The comparison between BGs and MD/MBAR focuses primarily on the number of energy evaluations. While informative, this metric alone is incomplete. BGs rely on expensive neural network evaluations, whereas classical potentials such as Lennard–Jones or mW are extremely cheap. A comparison in terms of actual wall-clock time or GPU hours would provide a more realistic assessment of the computational trade-offs involved.

We thank the reviewer for this important point and agree that the number of energy evaluations alone does not fully capture the computational cost. For inexpensive empirical potentials, such as those studied here, free energy calculations using MD+MBAR can have a lower total computational cost than training and evaluating a flow model, despite the larger number of energy evaluations required. However, for commonly used potential energy functions, such as machine-learned interatomic potentials (MLIPs), the computational cost is largely dominated by energy evaluations, making their number the primary factor to consider.

In the following, we illustrate how the computational costs of these two approaches differ for empirical potentials and for MLIPs. For reference, the time per MD step using the mW potential was measured in this work to be approximately 10^{-7} s per atom, while a representative MLIP evaluation is assumed to require around 10^{-4} s per atom [S6]. While these timings include both energy and force evaluations, which are required for flow training and MD simulations, force calculations are not needed for MBAR or flow inference. We neglect this difference here and assume identical timings for energy evaluations in MBAR and during inference, as this contribution is small compared to the overall computational cost.

Table 1 reports estimated the wall clock times required for training and inference with the flow model, as well as for a standard MD+MBAR calculation. The values for the flow model are based on $6 \cdot 10^7$ energy evaluations to train the model on a small system with $N = 216$ and $5 \cdot 10^4$ evaluations for inference on the large system with $N = 1728$ using the mW potential, as reported in the manuscript. The model overhead for both training and inference was computed by subtracting the time spent on energy evaluations from the total training and inference times, and is assumed to be independent of the potential. The number of energy evaluations for an MD+MBAR calculation is similarly assumed to be $6 \cdot 10^7$. The MLIP timings shown in Table R1 are estimated under the assumption that the number

of training steps is comparable to that used for the empirical potential.

Table R1: Estimated wall-clock times (in hours) for flow-based and MD+MBAR free energy calculations using the mW potential and a representative MLIP. Training uses $N = 216$ particles, while inference uses $N = 1728$ particles. Times are split into energy evaluations and model/optimization overhead where applicable.

Stage	mW (h)	MLIP (h)
Flow Training ($N = 216$): Energy evals	0.4	360.0
Flow Training ($N = 216$): Model overhead	191.6	191.6
Flow Inference ($N = 1728$): Energy evals	0.002	2.4
Flow Inference ($N = 1728$): Model overhead	0.2	0.2
Flow Total ($N = 1728$)	192.2	554.2
MD+MBAR ($N = 1728$): Energy evals	2.9	2880.0

As can be seen from Table 1, for inexpensive empirical potentials, MD+MBAR is significantly cheaper than the flow-based approach, with a total cost of 2.9 h compared to 192.2 h for the flow model at $N = 1728$. However, this situation changes dramatically for MLIPs, where the flow-based approach requires 554.2 h, while MD+MBAR reaches 2880.0 h. This highlights that the number of energy evaluations becomes the dominant factor in the overall computational cost. Importantly, the runtime of the flow-based method is largely dominated by the training phase, which can be further amortized through conditional training, allowing the model to generalize across external parameters and thereby reducing the cost of subsequent free energy calculations.

We have added the following paragraph to the main text to discuss the trade-offs between computational cost and model efficiency for different interaction potentials:

“While BG models are highly efficient compared to MD+MBAR in terms of the number of required energy evaluations, this advantage does not directly translate to wall-clock times for a single free energy estimate in the model systems studied here. For the cubic ice system with $N = 1728$, a full MD+MBAR free energy estimate only takes a few hours, including simulations at all intermediate states and the MBAR evaluation, whereas training the flow model to convergence required approximately 8 GPU days (see Supplementary Note F for details). However, the comparison of wall-clock times depends strongly on the cost of the energy evaluations which for inexpensive interaction models, such as mW, is on the order of 10^{-7} s per atom. Machine-learning interatomic potentials, now widely regarded as the workhorse of computational materials science, are considerably more expensive than empirical potentials, typically by a factor of 10^3 to 10^4 [50]. Here, energy evaluations dominate the computational cost, while the overhead from optimization and neural network evaluation remains largely independent of the chosen interaction model. Consequently, the number of energy evaluations becomes the most important factor and the reduced number required by the BG approach is expected to result in relative wall-clock times that are considerably more favorable or even significantly reduced compared to conventional MD+MBAR calculations.”

4. The paper computes Gibbs free energies using a Legendre transform from Helmholtz free energies. To validate that this procedure is sufficiently accurate, I suggest computing thermodynamic response functions such as the isothermal compressibility from the Gibbs free energy and comparing them with reference values from MD. This would provide a stringent and physically meaningful test of the proposed framework.

We thank the reviewer for this helpful suggestion. To validate the accuracy of the Legendre transform from Helmholtz to Gibbs free energies, we computed the isothermal compressibility directly from the

resulting Gibbs free energy. From the Gibbs free energy, the volume follows as

$$V = \left(\frac{\partial G}{\partial P} \right)_T, \quad (\text{R4})$$

and the isothermal compressibility is obtained from the second pressure derivative,

$$\kappa_T = -\frac{1}{V} \left(\frac{\partial V}{\partial P} \right)_T = -\frac{1}{V} \left(\frac{\partial^2 G}{\partial P^2} \right)_T. \quad (\text{R5})$$

In practice, we evaluated $V(P)$ and $\kappa_T(P)$ for cubic silicon ($N = 216$) from numerical derivatives of $G(P)$ computed on a grid of 25 pressure points spanning the range 10 GPa to 15 GPa. To obtain stable higher-order derivatives, the $G(P)$ data were smoothed using a local polynomial (Savitzky–Golay) filter prior to differentiation, and the resulting $V(P)$ curve was additionally fitted with a Murnaghan equation of state to provide a robust estimate of the pressure derivatives. We then compared these compressibilities with independent reference values obtained from an MBAR calculation, in which $G(P)$ was computed along the same set of 25 pressure points and processed using the identical analysis procedure, as well as from direct molecular dynamics simulations in the NPT ensemble, where the isothermal compressibility is given by the fluctuation formula

$$\kappa_T = \frac{\langle V^2 \rangle - \langle V \rangle^2}{k_B T \langle V \rangle}. \quad (\text{R6})$$

As shown in Fig. R1, the compressibility derived from the Gibbs free energy shows very good agreement with the MD reference data across the investigated pressure range, both in absolute magnitude and in its pressure dependence. In particular, the low-pressure values and the expected decrease of κ_T with increasing pressure are reproduced within the reported uncertainties.

A corresponding discussion of the calculation of derived quantities has been added to Supplementary Note D of the revised SI and we mention the possibility of computing derived quantities on page 6 of the revised manuscript:

“The Gibbs free energies computed in this manner also enable the calculation of derived quantities, such as isothermal compressibilities (see Supplementary Notes C and D for details on the conditional training and derived quantities, respectively).”

Minor Revision Remarks:

1. In the target distribution, the auxiliary variables (a) are modeled as Gaussian-distributed around the physical coordinates (x), i.e. $p(a|x) = \mathcal{N}(a; x, \eta^2 I)$. While this construction is standard in augmented flow frameworks, it would be good to explain here why this particular design is chosen and what intuition it provides.

We thank the reviewer for this comment. We have revised the formulation so that the intuition behind modeling the auxiliary variables as Gaussian-distributed around the physical coordinates is clearer, with the auxiliary system retaining structural information that can be exploited by the flow transformation. Additionally, we have moved the following paragraph rationalizing the design of the flow transformation (previously Eq. 4, in the revised manuscript Eq. 2) for the augmented flow immediately after introducing the joint distribution, so that the advantages of this approach over standard coupling flows are more directly evident:

“Utilizing the auxiliary variables, we define the flow transformation for particle i by

$$\mathbf{x}'_i = g(\mathbf{x}_i | \mathbf{h}_i^a) \quad , \quad (\text{2})$$

Figure R1: Isothermal compressibility κ_T of cubic silicon as a function of pressure obtained from three independent estimators: (i) derivatives of the BG-derived Gibbs free energy combined with a Murnaghan equation-of-state fit to the corresponding molar volume $V(P)$, (ii) derivatives of the MBAR Gibbs free energy $G(P)$ processed using the same smoothing and equation-of-state fitting procedure, and (iii) volume fluctuations from MD simulations. Solid lines denote the mean values, while shaded bands indicate the corresponding uncertainties which were obtained over five runs.

where g is parametrized by the embedding \mathbf{h}_i^a , which is computed using a graph neural network (GNN) [26] that aggregates information from each auxiliary particle’s local neighborhood (see Fig. 1). Details on the implementation can be found in Supplementary Note C. A crucial feature of the local approach presented here is its linear scaling with system size, achieved by fixing the number of neighbors. This stands in stark contrast to global architectures based on, for example, attention mechanisms which learn interactions between all particles, resulting in quadratic scaling with system size [40, 15].”

2. The marginal distribution over physical coordinates is obtained by integrating out the auxiliary variables. Currently, the derivation is deferred almost entirely to the Supplementary Information. Given its central importance, the authors should explicitly outline the key steps of the integration.

We thank the reviewer for this suggestion. As the section is short, we have moved the derivation from the Supplementary Information to the main text, explicitly outlining the key steps of integrating out the auxiliary variables. Given the importance of the marginal distribution for the developed approach, we have further explicitly studied the convergence of the marginal-based free energy estimates in Supplementary Note A.

3. η_q in $q(x) = \mathcal{N}(x; 0, \eta_q^2 \mathbf{I})$ is not defined.

We thank the reviewer for this observation. We have clarified in the manuscript that $\eta_q \mathbf{I}$ represents the covariance matrix of the physical base distribution, where $\eta_q \in \mathbb{R}_{>0}$ is the standard deviation.

References

- [S1] Jonas Köhler et al. “Rigid Body Flows for Sampling Molecular Crystal Structures”. In: *Proceedings of the 40th International Conference on Machine Learning (ICML)*. Vol. 202. 2023, pp. 17301–17326. URL: <https://proceedings.mlr.press/v202/kohler23a.html>.

- [S2] E. Olehnovics et al. "Lattice free energies of molecular crystals using normalizing flow". In: *ChemRxiv* (2025). DOI: 10.26434/chemrxiv-2025-92x2f.
- [S3] Claudio Zeni et al. "A generative model for inorganic materials design". In: *Nature* (2025). DOI: 10.1038/s41586-025-08628-5.
- [S4] Benjamin Kurt Miller et al. "FlowMM: Generating Materials with Riemannian Flow Matching". In: *Forty-first International Conference on Machine Learning*. 2024. URL: <https://openreview.net/forum?id=W4pB7VbzZI>.
- [S5] Cheng Zeng et al. *MolCrystalFlow: Molecular Crystal Structure Prediction via Flow Matching*. 2026. arXiv: 2602.16020 [cs.LG]. URL: <https://arxiv.org/abs/2602.16020>.
- [S6] Ryan Jacobs et al. "A practical guide to machine learning interatomic potentials – Status and future". In: *Curr. Opin. Solid State Mater. Sci.* 35 (Mar. 2025), p. 101214. ISSN: 1359-0286. DOI: 10.1016/j.cossms.2025.101214. URL: <http://dx.doi.org/10.1016/j.cossms.2025.101214>.

-----Reply to the report of the second reviewer -----

We thank the reviewer for the thorough assessment of our manuscript. In order to address the points raised, we paste and quote the referee report and insert our reply.

This manuscript presents a substantial and timely advance in the development of Boltzmann Generators for the equilibrium sampling of condensed-phase materials. By introducing a local, size-transferable augmented flow architecture based on graph neural networks, the authors convincingly overcome one of the main limitations of prior BG approaches: their poor scalability and prohibitive training costs for large systems. The paper is technically sound, carefully executed, and, in my view, represents a significant contribution to the field of machine-learning-based molecular simulation.

Formulating the generative model in terms of local environments while retaining exact likelihoods through augmented coupling flows addresses a significant bottleneck in the current literature. The resulting architecture demonstrates transferability across system sizes, enabling models trained on relatively small cells to be applied to systems with over a thousand atoms. The numerical evidence supporting this claim is compelling. Structural observables, energy distributions, effective sample sizes, and free-energy estimates are validated against high-quality MD+MBAR reference calculations, showing excellent agreement across multiple model systems, including Lennard-Jones crystals, monatomic water ice, and Stillinger–Weber solids.

The demonstration that accurate absolute and relative free energies can be obtained for large crystalline systems at a computational cost competitive with, and in some cases superior to, state-of-the-art free-energy methods is particularly noteworthy. The discussion of size convergence and finite-size effects is thorough and convincing. Application to phase stability problems, such as FCC–HCP Lennard-Jones, ice polymorphs, and the silicon phase diagram, underscores the practical relevance of the approach. Conditioning on thermodynamic variables and interaction parameters further strengthens the study by illustrating how training costs can be amortized across state points and materials classes.

The manuscript is generally clear and well structured, and the figures are informative and thoughtfully designed. Any remaining weaknesses are minor.

We thank the reviewer for the positive feedback and for recognizing the technical soundness, careful execution, and contribution of our work. We have addressed the minor suggestions and clarified the relevant points in the revised manuscript, correspondingly.

For example, a slightly more explicit discussion of the limits of the locality assumption and the observed variability in sampling efficiency across different crystal structures could help guide future applications, where the number of degrees of freedom might scale with both chemical complexity and system size.

The locality assumption underlying the present approach is expected to break down in systems with significant long-range interactions, such as electrostatics, where distant degrees of freedom contribute non-negligibly to the change in local configurations. While the generation of samples would be affected by this, the evaluation of the energy depends on the potential of choice. Thus, if the local environment is too small, it should become visible. To emphasize the limitations of a purely local formulation in the context of long-range interactions we have added the following sentence to the discussion:

“Another important direction for future work is the development of models capable of capturing long-

range interactions, such as those arising from electrostatics, which cannot be represented within any purely local formulation.”

The variations in observed sampling efficiency for different systems is currently purely empirical but not limited to our approach. For instance, mW ice phases have been found to train in general more easily as compared to LJ phases which might be related to the nature of the bonding (the LJ potential is isotropic whereas the mW potential promotes more directional bonds). These types of effects are certainly of interest in future studies. To nevertheless make the reader aware of these aspects, we have added the following the discussion:

“Future directions include extending this approach to efficiently sample liquid phases and developing models with broad generalization across chemical space. In this context, it will be important to investigate why the models exhibit different performances for different combinations of potentials and structures, to better understand and guide the design of generative approaches.”

Similarly, clarifying under which conditions accurate free-energy estimates remain possible despite very small effective sample sizes would benefit readers less familiar with targeted free-energy perturbation. These points are primarily matters of exposition and do not detract from the strength of the results.

Accurate free-energy estimates can in principle still be obtained even when the effective sample size is small, provided that sufficient overlap exists between the relevant ensembles. In practice, however, the ESS becomes extremely noisy at small values (see Table 1 of the main text), which limits its usefulness as a quantitative reliability metric in this regime. Consequently, caution is warranted when interpreting estimates obtained from extremely small ESS, and additional validation (e.g., consistency across independent runs or alternative estimators) may be necessary. To emphasize that caution is warranted when evaluating the reliability at low ESS, we have added the following to the discussion of accuracy:

“For the largest system size ($N = 1728$), the joint estimates slightly overestimate the true values, while the marginal evaluation partially corrects for this, yielding more accurate results in agreement with the ESS reported in Tab. 1. In this context, it is worth mentioning that it is difficult to verify the reliability of a given free energy estimate at low ESS, since the variance of the estimates can remain very small even when ESS is low (see Supplementary Note E). Based on the examples considered, a mean ESS of 0.1% or higher appears to be a sufficient criterion to ensure deviations in the free energy estimates remain below $10^{-3}k_B T$ per particle when using $5 \cdot 10^4$ samples. A similar behavior was observed for the FCC LJ crystal, with the corresponding free energy estimates reported in the Supplementary Note E.”

In summary, this is a high-quality, impactful study that extends the practical applicability of Boltzmann Generators well beyond previous work. I strongly recommend publication in its present form, with only minor optional clarifications at the authors' discretion.

We thank the reviewer for the very positive evaluation and the recommendation for publication. We are pleased that the reviewer recognizes the significance and impact of our work.

-----Reply to the report of the third reviewer -----

We thank the reviewer for the detailed assessment of our manuscript. In order to address the points raised, we paste and quote the referee report and insert our reply.

The present work introduces an architecture of normalizing flows (NFs) that are transferable in size for the purpose of studying crystals. The proposed architecture resembles the previous proposition of Ref [39]. However, here the authors restrict to a local neighborhood of each particle in the update rule at each layer. The authors show that this locality, implemented with a graph neural network, allows to train the model for a certain crystal size and then use it to sample a larger crystal, typically 5 times larger in their experiments. Training can be performed by optimizing the reverse KL in this application where the goal is to model the distribution close-by the perfect crystal and no multimodality is expected.

Despite using an augmented state space, that precludes exact likelihood computation of the model on the physical space, the authors describe means of evaluation of free energy differences through targeted free energy perturbation and report results that appear stable either when the marginal likelihood is estimated through a Monte Carlo or when relying on the free energy difference on the augmented space which should be equal to the marginal one (given the chosen joint law). Results are reported for cubic and hexagonal ice, FCC and HCP Lennard Jones and the Stillinger-Weber tunable potential with crystal sizes up to the order of 1000 particles. The NVT and the NPT ensembles can both be studied, in the latter case by training NFs conditional on the shape of the crystal cell. Experiments are sound and thorough, discussing various aspects of the method.

As expected the efficiency of sampling degrades as the size of crystals increases dropping to 0.02% for 1500 particle crystals. This typical size appears to be just enough to converge the free energy estimates. Variability of the results for different crystals calls for more systematic assessment as it seems plausible that the method will fail in some cases (requiring larger cells). However, the comparison in terms of number of calls to the force compared to MD+MBAR is interesting, showing that the proposed method is not drastically cheaper when taking into account the training cost, but that once training is done the free energy computations are cheaper in the considered examples (reaching ESS of 0.02% again) and all the more so if amortizing the training across some thermodynamic parameter.

The paper is overall well written, it describes a new methodology which is demonstrated in various experiments. The limits are honestly discussed in the text if not in the introduction and conclusion. I am in favor of accepting the paper while I note that further research will be needed to fully understand the potential of the method given the differences of performance observed for different crystals. A few clarifications would still further improve the manuscript:

We thank the reviewer for the thorough and positive assessment of our work and the clear recommendation for publication in *Nature Communications*. We are pleased that the reviewer recognizes the novelty and practical relevance of our approach. We also appreciate the constructive suggestions for clarifications, and we have addressed them in the revised manuscript to improve readability and completeness.

1. As far as I could follow the motivation for using augmented architectures in Ref [39] was to define an NF with invariant distribution with respect to some symmetries of the space. Here, the authors do not seem to be imposing any invariance (beyond the permutation invariance implied by the GNN). Could the authors clarify was they resort to augmented flows here? Making the following sentence more explicit may help "As coupling flows need to split coordinates either across particles or across

spatial dimensions or a combination thereof, the calculation of full three-dimensional environments is prohibited [38]". Have they compared with the performance of traditional coupling flows?

We thank the reviewer for this insightful question. In Ref. [39], a key requirement for constructing SE(3)-equivariant flow models is the ability to compute distances and relative displacement vectors within the architecture. While equivariance is not required in our setting, the ability to compute distances remains essential, as our local formulation likewise depends on relative geometric information. Standard coupling flows do not provide access to such distance or difference-vector computations due to the required splitting of the full coordinates, which motivates the use of an augmented state space in our approach.

We have clarified the corresponding sentence about limitations of standard coupling flows in the main text and added an illustrative example in Supplementary Note A to further explain why standard coupling flows cannot incorporate full three-dimensional local environments. Additionally, we have moved the following paragraph rationalizing the design of the flow transformation (previously Eq. 4, in the revised manuscript Eq. 2) for the augmented flow immediately after introducing the joint distribution, so that the advantages of this approach over standard coupling flows are more directly evident:

"Utilizing the auxiliary variables, we define the flow transformation for particle i by

$$\mathbf{x}'_i = g(\mathbf{x}_i | \mathbf{h}_i^a) \quad (2)$$

where g is parametrized by the embedding \mathbf{h}_i^a , which is computed using a graph neural network (GNN) [26] that aggregates information from each auxiliary particle's local neighborhood (see Fig. 1). Details on the implementation can be found in Supplementary Note C. A crucial feature of the local approach presented here is its linear scaling with system size, achieved by fixing the number of neighbors. This stands in stark contrast to global architectures based on, for example, attention mechanisms which learn interactions between all particles, resulting in quadratic scaling with system size [40, 15]."

Regarding a comparison with traditional coupling flows, this is already included and an important aspect of our work. In the manuscript, we refer to traditional coupling flow as *global* flows as compared to the introduced *local* flows. In particular, we evaluated the Transformer-based architecture used in Refs. [15] and [17], and explicitly compare the results in Fig. 3 and Table 1 of the manuscript. These comparisons confirm the advantages of our local, augmented flow approach in terms of efficiency, accuracy, and, most importantly, transferability for crystalline systems.

2. The definition of Z_{aux} should be formally stated as it is a matter of convention. The most natural way is to define it as the partition of $a|x$ given the weight (S1) in SI. This yields $Z_{\text{aux}}(x)$, function of x , which indeed turns out to be independent of x consistently with the authors description.

We thank the reviewer for this comment. We have now formally defined $Z_{\text{aux}}(x)$ in Supplementary Note A and clarified that it is indeed independent of x due to the properties of the normal distribution.

3. In Figure 2, for the histogram of potentials for FCC LJ, the x axis could be better adapted so the agreement between BG and MD could be more finely assessed. There is a strange zoom out leaving significant blank space hurting the readability of the plot.

We thank the reviewer for this observation. We have adapted the x -axis limits in Figure 2 to improve readability.

4. "For both systems, the RDFs produced by the BGs are nearly indistinguishable from those obtained

via MD over length scales going far beyond the size of the training systems.” p5 sec 3.1, I must be misunderstanding something, the plot seems to report $g(r)$ for up to L_{train} , meaning “up to” the size of the training system and not “far beyond”?

We thank the reviewer for highlighting this potential source of confusion. For a given system, the radial distribution function (RDF) is only meaningful up to half of the minimum box dimension, i.e., $L/2$ in a cubic box, due to periodic boundary conditions. Consequently, the largest distance for which the RDF can be evaluated for the training system is $L_{\text{train}}/2$. For the systems to which the model is transferred, the box is exactly 8 times larger in total (i.e., 2 times larger per dimension), so that $L_{\text{transfer}}/2 = L_{\text{train}}$, which corresponds to the maximum r/σ value shown in the figure. We have also adjusted the figure caption to clarify this and reduce potential confusion:

“The red line indicates half the box length of the training system, which is the largest distance for which the RDF can be evaluated for this system. For the transferred systems, the RDF is similarly evaluated up to their corresponding half-box length.”

5. For figure 6: why the BCCs have lower numbers of particles than other phases?

We thank the reviewer for this question. Different crystal structures have different numbers of atoms in their unit cell, which means that not every cell can be composed into supercells of the same number of particles. The unit cells relevant to this work are detailed in Supplementary Note B of the revised SI. For the relevant phases of the SW potential, β -tin has 4 atoms per unit cell, cubic has 8, and BCC has 2. Based on these unit cells, supercells can be constructed to yield a comparable number of atoms across different phases. For example, $3 \times 3 \times 3$ and $3 \times 3 \times 6$ supercells can be built for the cubic and β -tin phases (both yielding 216 atoms), respectively, whereas a $4 \times 4 \times 4$ BCC supercell contains 128 atoms.

Remarks on code availability:

I have not run the code, but it features a readme and a tutorial notebook, functions appear to also be minimally commented.

-----Reply to the report of the fourth reviewer -----

We thank the reviewer for the careful assessment of our manuscript. In order to address the points raised, we paste and quote the referee report and insert our reply.

This paper addresses the scalability of Boltzmann generators. The method presented here allows for sampling of systems with more than a thousand atoms. Applied examples demonstrate the effectiveness of this approach.

The paper is well-written and it deserves to be published in Nature. Comm.

We thank the reviewer for the very positive assessment and for recognizing the significance and scalability of our approach. We appreciate the support for publication in *Nature Communications*.

Minor points:

- Introduction: The authors might want to avoid defining their own work as "seminal."

We thank the reviewer for this suggestion and have removed the term "seminal".

-The systems presented here are far smaller than most systems simulated by MD/MC, such as biological systems. What are the prospects for sampling the latter using this method?

We thank the reviewer for raising this point. While our method shares some conceptual similarities with architectures employed for modeling biological systems, the approach developed here is specifically tailored to crystalline materials. In particular, modeling only the displacements from a reference lattice is especially useful for crystalline materials. One could imagine applying a similar strategy to proteins, for example by modeling the displacements of sidechain atoms relative to the backbone. Nonetheless, materials provide a particularly suitable testbed for this approach due to their high spatial regularity, which distinguishes them from proteins. In fact, state-of-the-art Boltzmann Generators for proteins are currently limited to modeling relatively small peptides (see e.g. Ref. [S1]).

- Some sentences seem redundant and could be removed without altering the scientific content. For example, "The presented results highlight the potential of our architecture to serve as a powerful and scalable tool for studying complex materials across a wide range of thermodynamic conditions."

We thank the reviewer for this comment. We have reviewed the manuscript carefully and believe that the sentence highlighted helps emphasize the practical relevance of our results, so we have retained it.

References

[S1] Charlie B. Tan et al. *Scalable Equilibrium Sampling with Sequential Boltzmann Generators*. 2025. eprint: arXiv:2502.18462.